# Synovial Fluid-Derived Extracellular Vesicles of Patients with Arthritides Contribute to Hippocampal Synaptic Dysfunctions and Increase with Mood Disorders Severity in Humans

**DOI:** 10.3390/cells11152276

**Published:** 2022-07-23

**Authors:** Clara Cambria, Francesca Ingegnoli, Eleonora Borzi, Laura Cantone, Lavinia Agra Coletto, Alessandra Stefania Rizzuto, Orazio De Lucia, Sabrina Briguglio, Massimiliano Ruscica, Roberto Caporali, Valentina Bollati, Massimiliano Buoli, Flavia Antonucci

**Affiliations:** 1Department of Medical Biotechnology and Translational Medicine (BIOMETRA), University of Milan, 20100 Milan, Italy; clara.cambria@unimi.it (C.C.); eleonora.borzi@gmail.com (E.B.); sabrina.briguglio@unimi.it (S.B.); 2Division of Clinical Rheumatology, ASST Gaetano Pini-CTO, 20100 Milano, Italy; francesca.ingegnoli@unimi.it (F.I.); lavinia.coletto@unimi.it (L.A.C.); orazio.delucia@asst-pini-cto.it (O.D.L.); roberto.caporali@unimi.it (R.C.); 3Department of Clinical Sciences & Community Health, Research Center for Adult and Pediatric Rheumatic Diseases, University of Milan, 20100 Milan, Italy; 4Department of Clinical Sciences and Community Health, University of Milan, 20100 Milan, Italy; laura.cantone@unimi.it (L.C.); valentina.bollati@unimi.it (V.B.); 5Department of Pharmacological and Biomolecular Science, University of Milan, 20100 Milan, Italy; alessandra.rizzuto@unimi.it (A.S.R.); massimiliano.ruscica@unimi.it (M.R.); 6Department of Neurosciences and Mental Health, Fondazione IRCCS Ca’ Granda Ospedale Maggiore Policlinico, Via F. Sforza 35, 20122, Milan, Italy; 7Department of Pathophysiology and Transplantation, University of Milan, 20100 Milan, Italy; 8Institute of Neuroscience, IN-CNR, 20100 Milano, Italy

**Keywords:** EVs, osteoarthritis, rheumatoid arthritis, synaptic transmission, neurons

## Abstract

Arthritides are a highly heterogeneous group of disorders that include two major clinical entities, localized joint disorders such as osteoarthritis (OA) and systemic autoimmune-driven diseases such as rheumatoid arthritis (RA). Arthritides are characterized by chronic debilitating musculoskeletal conditions and systemic chronic inflammation. Poor mental health is also one of the most common comorbidities of arthritides. Depressive symptoms which are most prevalent, negatively impact patient global assessment diminishing the probability of achieving the target of clinical remission. Here, we investigated new insights into mechanisms that link different joint disorders to poor mental health, and to this issue, we explored the action of the synovial fluid-derived extracellular vesicles (EVs) on neuronal function. Our data show that the exposure of neurons to different concentrations of EVs derived from both RA and OA synovial fluids (RA-EVs and OA-EVs) leads to increased excitatory synaptic transmission but acts on specific modifications on excitatory or inhibitory synapses, as evidenced by electrophysiological and confocal experiments carried out in hippocampal cultures. The treatment of neurons with EVs membrane is also responsible for generating similar effects to those found with intact EVs suggesting that changes in neuronal ability arise upon EVs membrane molecules′ interactions with neurons. In humans with arthritides, we found that nearly half of patients (37.5%) showed clinically significant psychiatric symptoms (CGIs score ≥ 3), and at least mild anxiety (HAM-A ≥ 7) or depression (MADRS and HAM-D ≥ 7); interestingly, these individuals revealed an increased concentration of synovial EVs. In conclusion, our data showing opposite changes at the excitatory and inhibitory levels in neurons treated with OA- and RA-EVs, lay the scientific basis for personalized medicine in OA and RA patients, and identify EVs as new potential actionable biomarkers in patients with OA/RA with poor mental health.

## 1. Introduction

The arthritides are a highly heterogeneous group of disorders in which synovial joints represent the primary target. They include two major clinical entities, localized joint disorders such as osteoarthritis (OA), and systemic autoimmune-driven diseases such as rheumatoid arthritis (RA) and spondyloarthritis (SpA). Despite considerable differences in clinical and pathogenetic features, arthritides are characterized by a significant burden that impacts patients’ quality of life. OA is the most prevalent and chronic debilitating musculoskeletal condition affecting about 7% of the population worldwide [1], while RA and SpA are the most common systemic inflammatory diseases with a prevalence of up to 2% worldwide [2,3]. They both affect the whole joint structure but in a different manner, OA with cartilage degradation, osteophytes formation, subchondral bone remodeling, and synovitis [4], while RA leads to synovium proliferation, cartilage destruction, and bone erosions [5]. Thus, underpinning their joint involvement and chronic joint pain are different genetic, structural, mechanical, immunologic, metabolic, and inflammatory pathways involved in their pathogenesis. Moreover, the arthritides disease burden is worsened by the co-existence of diverse conditions (comorbidities) with their complications that accrue after diagnosis [6,7]. Accumulating evidence suggests that poor mental health is one of the most common comorbidities of arthritides, and both are major contributors to global disability; these conditions are intertwined and negatively affect their mutual course [8,9,10,11]. For example, the presence of depressive symptoms is associated with numerous deleterious outcomes such as increased mortality, work disability, worsened disease activity, and physical function, higher pain levels, and fatigue [12]. Moreover, depressive symptoms impact negatively on patient global assessment thus diminishing the probability of achieving the target of clinical remission [13,14].

New insights into mechanisms that link different joint disorders to poor mental health are needed to identify new potential actionable biomarkers to drive more personalized therapeutic strategies.

The first step is to look into how peripheral inflammation is spread into the central nervous system (CNS), and the majority of insights originate from animal models [15,16,17].

Peripheral inflammatory mediators can communicate with CNS either by direct routes, activating peripheral nociceptive afferents [18] and autonomic nervous system (ANS) [19], or by indirect routes such as circumventricular organs by volume diffusion, the choroid plexus, both physiologically and pathologically sites of increased blood-brain barrier (BBB) permeability [15], and possibly by active transport or tight junction damage [20,21]. Moreover, BBB, instead of being only a passive barrier, may actively undergo non-disruptive alterations due to peripheral inflammation and react by upregulating adhesion markers, activating endothelial cells, in turn, able to secrete inflammatory mediators [22,23]. Endothelial dysfunction is well-documented in RA, partially explaining the higher incidence of cardiovascular diseases [24], but it might also contribute to the neuropsychiatric symptoms associated with RA [25].

It remains to further elucidate the neuroinflammation process, meaning how CNS astrocytes, microglia, and macrophages acquire an inflammatory phenotype after receiving the aforementioned peripheral signals in the context of supraphysiological amounts of pro-inflammatory cytokines in the brain. Starting from peripheral joint inflammation, neuroinflammation establishes and translates into impaired neuronal function, hence neuropsychiatric symptoms. Among the proposed mechanisms already identified there are (i) the impairment of neurotransmitter (glutamate, serotonin, dopamine) metabolism and signaling by chronic inflammation evinced both in murine and in humans [26,27,28]; (ii) the dysregulation of hypothalamic-pituitary-adrenal (HPA) axis and cortisol production with consequent altered responses to stress [29]; (iii) changes in neuronal structure in terms of remodeling dendritic spines in murine models [30] or different patterns of brain connectivity in RA patients evaluated by functional and structural MRI [31]; and (iv) impaired hippocampal neurogenesis in lymphoid based murine models [23] or in RA patients [32], mirrored by lower levels of brain-derived neurotrophic factors [25] in brain arthritic rats [33]. What has been only marginally addressed but extremely relevant is the role of extracellular vesicles (EVs) released from the inflamed periphery in the direct control of neuron and glia functions. EVs are a heterogeneous group (40–500 nm) of cell-derived bilayer structures secreted by nearly all cells into the extracellular space and loaded with proteins, lipids, and nucleic acids [34,35]. They are key players in both the short- and long-range intercellular communication in physiological and pathological conditions by transferring a molecular message encoded by their cargo or membrane components to recipient cells. EVs are found in body fluids such as cerebrospinal fluid (CSF), blood, urine, saliva, and synovial fluids. In the brain, they are released from all types of cells, i.e. neurons, astrocytes, microglia, and oligodendrocytes [36] under different conditions of activation thus participating in intercellular communication, myelination, regulation of synaptic plasticity, and guaranteeing trophic support to neurons [37]. Neurons and glia cells can interact with their neighboring cells via the exchange of EVs, as an example, EVs released from activated microglia in the brain regulate and eventually propagate neuroinflammation by releasing proinflammatory cytokines (IL1-β, TNF-α) [38] and promoting neuronal excitation in the network [39,40]. In this context, systemic inflammation would generate neuroinflammation and lastly neuronal dysfunctions (i.e. accounting for vulnerability to mental conditions) by different pathways, i.e. by (i) the direct action of cytokines coming from the periphery into neurons (as mentioned above) and (ii) persistent high cytokines levels generated by glia-derived EVs. On the contrary, the direct impact of EVs coming from the inflamed peripheries to neurons is much less investigated. Indeed, only recently it has been shown that (i) pro-inflammatory EVs from the periphery have the potential to induce inflammation in the brain as revealed by ELISA quantification of TNF and IL-1B levels, and (ii) the ability of peripheral EVs to pass the BBB [41,42,43].

In light of this, EVs have been suggested as a potential link between peripheral joint inflammation and poor mental health. Thus, we isolated and characterized EVs from synovial fluids to evaluate how and to which extent they are able to influence mood disorders, and change neuronal transmission; an issue never explored before (see graphical abstract).

## 2. Methods

### 2.1. Design and Participants

This cross-sectional study was approved by the local ethics committee “Comitato Etico Milano Area 2” (approval code: 591_2020 bis). The study population consisted of all consecutive patients (aged > 18 years) with RA [44,45], SpA [46,47], or primary OA referred to the Division of Clinical Rheumatology, ASST Pini-CTO for aspiration of joint effusion and subsequent drug injection, as part of the therapeutic procedures in clinical practice. After signing the informed consent, the subject’s eligibility was verified, data was collected and coded.

### 2.2. Data Collection

At the time of arthrocentesis, for each patient, the following data were collected: (1) demographic and clinical variables; (2) rating scales for mental health; (3) ultrasonographic images. During the procedure for aspiration of joint effusion, discarded synovial fluid was collected and used for the analysis and experiments described below.

Mental health was evaluated by a clinical interview, and administration of rating scales to assess the presence and severity of affective symptoms and specific personality traits (including autistic ones): Montgomery Asberg Depression Rating Scale (MADRS), Hamilton Depression Rating Scale (HAM-D) 21 items, Hamilton Anxiety Rating Scale (HAM-A), Clinical Global Impression-Severity of Illness (CGIs), 10-item Big Five Inventory (BFI), 10-item Autism Spectrum Quotient (AQ-10). All the scales were administered by trained collaborators with the exclusion of BFI and AQ 10 which are self-administered. MADRS is a tool used to assess the core depressive symptoms (e.g., anhedonia) [48]; HAM-D is more sensitive to detect somatic and anxiety symptoms associated with depression [49]; HAM-A is a tool that defines the severity of anxiety [50]; CGIs is a simple tool to evaluate global severity of illness [51]; BFI is a very short scale that measures the prominence of the main five personality traits (agreeableness, conscientiousness, neuroticism, extroversion, openness) [52] and finally AQ-10 is a very short questionnaire assessing the presence and prominence of autistic personality traits [53]. With regard to BFI, we focused particularly on neuroticism as the predominance of this personality trait was associated with poor mental health [54]. All the scales (for AQ the full version) were validated in Italian samples.

Ultrasound assessment of the same joint of arthrocentesis was performed. Joints were examined using an 18 MHz linear transducer (Esaote MyLab 70, Esaote Genoa, Genova, Italy). The findings were assessed by gray scale (GSUS) and power Doppler (PDUS) with the standard setting according to recommendations [55,56,57,58]. Grey scale was scored from 0 to 3 by synovial thickness as follows; score 0: absent (<2 mm), score 1: mild synovial hypertrophy (<5 mm), score 2: moderate synovial hypertrophy (≥5 mm), score 3: intense synovial hypertrophy (>8 mm) [56,57,58]. Pulse repetition frequency was 750 Hz, Doppler frequency 9.1 MHz, and Doppler gain to avoid random noise was used. PD signal, which characterizes vascularity in the synovial tissue, also scored from 0 to 3 as follows; score 0: normal (undetectable power Doppler vessel signals in ultrasonographic synovial thickening area), score 1: mild (intrasynovial power Doppler flow signal distribution was detectable over <25% of the synovial area), score 2: moderate (<50%), score 3: marked (>50%). The highest grade throughout the knee joint was adopted [56,57,58,59].

### 2.3. Enrichment and Characterization of Synovial Fluid EVs

EV enrichment from synovial fluid was performed by serial centrifugations, followed by the ultracentrifugation step. Subsequently, EVs were characterized by Nanoparticle Tracking Analysis (NTA), Western Blotting, Flow Cytometry, and Micro-BCA. All procedures were conducted in accordance with MISEV 2018 guidelines [60], as detailed below.

#### 2.3.1. Synovial Fluid Processing and EV Enrichment

Blood samples were collected in ethylenediamine tetra-acetic acid (EDTA) tubes and processed within 2 h from the phlebotomy as described below. For each subject, an aliquot of 3 mL of synovial liquid was centrifuged at 1000, 2000, and 3000× *g* for 15 min at 4 °C. After each centrifugation step, pellets were discarded to remove cell debris. The resulting supernatants were subjected to ultracentrifugation (UC) at 110,000× *g* for 75 min at 4 °C in polypropylene UC tubes (Beckman Coulter; Brea, CA, USA) filled with phosphate-buffered saline (PBS) previously filtered through a 0.10 μm pore-size polyethersulfone filter (StericupRVP, Merck Millipore; Burlington, MA, USA). The EV-rich pellet obtained by UC was then resuspended in 0.5 mL of triple-filtered PBS (pore size 0.1 µm) to perform NTA and flow cytometry analysis.

Experiments carried out using either the soluble component or the membrane pellet were preceded by disrupture of intact EVs by multiple freeze and thaw cycles: broken EVs were repelleted at 100,000× *g* for 60 min [40] and the membrane pellet or luminal content obtained were then resuspended in filtered PBS.

#### 2.3.2. Nanoparticle Tracking Analysis (NTA)

NTA analysis was carried out with the NanoSight NS300 system (Malvern Panalytical Ltd., Malvern, UK). For each sample, five 30 s records were registered. NTA output was then analyzed by integrated NTA software (Malvern Panalytical Ltd., Malvern, UK), providing high-resolution particle size distribution profiles and EV concentration measurements (expressed as 10^6^ for 1 mL of synovial liquid).

#### 2.3.3. Western Blotting

EVs were resuspended in RIPA buffer (0.05 M Tris-HCl pH 7.7, 0.15 M NaCl, 0.8% TritonX-100, 0.8% sodium deoxycholate, and 0.08% SDS, 10 mM EDTA, 100 μM sodium orthovanadate, 50 mM NaF, 5 mM iodoacetic acid) containing a cocktail of protease and phosphatase inhibitors (Sigma Aldrich, Milan, Italy). After 1 h on ice, supernatant was centrifuged at 14,000× *g* for 10 min. Protein concentration was determined by BCA protein assay. Twenty micrograms of proteins and a molecular mass marker (Novex Sharp Protein Standard, Invitrogen (Thermo Fisher Scientific), Whaltam MA, USA) were separated on 10% SDS-PAGE and were transferred to a nitrocellulose membrane at 200 mA for 120 min. After washing and blocking, the blot was incubated with a diluted solution of the human primary antibody Alix (1:1000; Abcam, Cambridge, UK) and APOA1 (1:1000; Calbiochem Millipore, Burlington, MA, USA). Membrane was washed and exposed for 90 min to a diluted solution of the anti-secondary antibody (1:1000 anti-rabbit; New England Biolabs, Ipswich, MA, USA). Immunoreactive bands were detected by exposing the membranes to Clarity Western ECL chemiluminescent substrates (Bio-Rad Laboratories, Milan, Italy) and images were acquired with a ChemiDoc XRS System (Bio-Rad Laboratories, Milan, Italy).

#### 2.3.4. Flow Cytometry

The cellular origin of synovial fluid EVs was determined by immunophenotyping using the MACSQuant Analyser flow cytometer (Miltenyi Biotec, Bergisch Gladbach, Germany, [61]), after calibration with Fluoresbrite Carboxylate Size Range Kit I (0.2, 0.5, 0.75, and 1 µm). To evaluate EV integrity, 60 µL sample aliquots were stained with 0.02 µM 5(6)-carboxyfluorescein diacetate N-succinimidyl ester (CFSE) at 37 °C for 20 min in the dark. CFSE is a vital dye that can passively enter EVs, where intracellular esterases remove the acetate group and convert the molecule into the fluorescent ester form. To characterize and count EVs, the following panel of antibodies conjugated with allophycocyanin (APC) was used: AbCD177-APC (neutrophils, clone REA 598; APC: allophycocyanin, Miltenyi Biotec), AbCD14-APC (monocytes, clone TUK4, Miltenyi Biotec, Bergisch Gladbach, Germany), AbCD62E-APC (activated epithelium, clone REA280, Miltenyi Biotec, Bergisch Gladbach, Germany), AbCD25-APC (T-reg cells, clone REA945, Miltenyi Biotec, Bergisch Gladbach, Germany). Before use, each antibody was centrifuged at 17,000× *g* for 30 min at 4 °C to remove aggregates. A triple-filtered PBS aliquot was stained with CFSE and the mentioned antibodies to evaluate auto-fluorescence and to determine the gating strategy. Quantitative multiparameter analysis of flow cytometry data (expressed as 10^3^ for 1 mL of plasma) was performed using FlowJo software (Tree Star, Inc., Ashland, OR, USA).

#### 2.3.5. Micro BCA

To evaluate the total amount of protein in each EVs samples, the Micro BCA Protein Assay Kit (ThermoFisher, Whaltam, MA, USA) was used as described in the manufacturer’s protocol. The assay plate was incubated at 37 °C for 30 min and then the absorbance was measured at 562 nm on a spectrophotometer plate reader (Victor2-1420 multilabel counter, Wallac, PerkinElmer Italia Spa, Milan, Italy). Once we obtained the protein concentration of each sample, we calculated the volumes of resuspended EVs to treat hippocampal neuronal cultures at three different concentrations: 0.75–2.4–4 μg/mL. Also, using the data obtained through NTA, we confirmed the increasing number of EVs (particles/mL) corresponding to the chosen concentrations.

### 2.4. Animals

The experimental procedures here performed were conducted in accordance with the guidelines provided by the Italian Council on Animal Care and were approved by the Italian Government Decree n° 27/2010 and by the Italian Legislation (L.D. n° 26/2014). All efforts were made in order to minimize the suffering and the number of animals used.

### 2.5. Cell Cultures

Primary hippocampal neurons were obtained from E18 wild-type mice embryos. Hippocampi collected were stored in HBSS 1× (10 mL HBSS 10X, GIBCO; 3.30 mL HEPES 0.3 M at pH 7.3; 1 mL PenStrep, Sigma Aldrich, Milan, Italy; 87 mL sterile water) and then chemically digested with 0.5% trypsin in HBSS 1× for 15 min at 37 °C. The tissues were gently dissociated with a sterile micropipette to isolate the cells. Neurons were plated on 24 mm slides previously coated with poly-L-lysine at the density of 150 k cell for each slide. Neuronal cultures were maintained in culture medium (100 mL Neurobasal, GIBCO; 2 mL B27 Supplement 50X, GIBCO; 1 mL PenStrep, Sigma Aldrich, Milan, Italy; 250 μL Glutamine 200 nM, GIBCO; 125 μL Glutamate 10 mM, Sigma Aldrich, Milan, Italy), in incubator at the constant temperature of 37 °C and in the presence of 5% CO_2_.

### 2.6. In Vitro Electrophysiology

Whole-cell recordings were performed in the voltage-clamp configuration on 14/15 DIV hippocampal neurons using an Axopatch 200 A amplifier and the pClamp-10 software (Axon Instruments, Forest City, CA, USA) as previously described [62,63]. Before recording, neuronal cultures were incubated for 2 h with vehicle (PBS) or with intact or broken EVs obtained from OA and RA patients at different concentration: 0.75 μg/mL; 2.4 μg/mL, and 4 μg/mL. Also, in the experiments with Etanercept (ETN) neuronal cultures were incubated for 2 h with EVs membrane alone (OA-EVs membrane at high concentration and RA-EVs membrane at low concentration) or in combination with ETN at the concentration of 200 ng/mL and subsequently, electrophysiological recordings were performed.

Recording pipettes were fabricated from glass capillary (World Precision Instruments) using a two-stage puller (Narishige, London, UK); they were filled with the intracellular solution Cs-gluconate (Cesium Gluconate 130 mM; CsCl 8 mM; NaCl 2 mM; HEPES 10 mM; EGTA 4 mM; MgATP 4 mM; GTP 0.3 mM; pH 7.3) and the tip resistance was 3–5 MΩ. Recordings were performed using as external solution KRH (Krebs’-Ringer’s-Hepes: 125 mM NaCl, 5 mM KCl, 1.2 mM MgSO4, 1.2 mM KH2PO4, 2 mM CaCl2; 6 mM glucose, 25 mM Hepes-NaOH pH 7.4). Neurons were held at holding potentials of −70 mV, to identify the excitatory miniature events, and at + 10 mV for the inhibitory miniature events. Currents were sampled at 10 kHz and filtered at 2 kHz. Analyses of the recorded traces have been performed using Clapmfit-pClamp 10 software; event threshold was set at −8 pA for mEPSCs and + 6 pA for mIPSCs. The E/I ratio was calculated as ratio of mEPSCs and mIPSCs frequencies measured in the same neuron.

### 2.7. Immunocytochemical Staining

Immunofluorescence staining was performed on 14/15 DIV hippocampal neurons incubated for 2 h with intact EVs obtained from OA and RA patients and then fixed in 4% paraformaldehyde and 4% sucrose for 8 min. Experiments were carried out using the following antibodies: rabbit anti-vGAT (1:1000, Synaptic System, Goettingen, Germany), guinea pig anti-vGLUT1 (1:1000, Synaptic System, Goettingen, Germany), mouse anti-βIII tubulin (1:250, Sigma). Secondary antibodies were conjugated with Alexa-488, Alexa-633 or Alexa-555 fluorophores (1:200) (Invitrogen, San Diego, CA, USA). Images were acquired using a LSM800 confocal microscope with 40X objective. Fluorescence images processing and analyses were performed with ImageJ Software (National Institutes of Health). It analyzed the density of vGAT and vGLUT1 as a ratio between the counted positive vGAT or vGLUT1 puncta over micron of dendrite. In addition, it also analyzed the size and the intensity of the puncta.

### 2.8. Statistics

With regard to clinical features (including mental health), descriptive analyses were performed. One-way analyses of variance were performed to compare rating scale scores between the two groups of patients (RA + SpA versus OA). Correlation analyses (Pearson’s r) were performed to investigate the relationship between body mass index (BMI) and rating scale scores.

The statistical analysis was performed using the Prism 8 software (GraphPad). Data were analyzed by unpaired parametric or nonparametric statistics: Student *t*-test or Mann–Whitney test were performed in presence of two experimental groups, otherwise, in the case of more groups, one-way analysis of variance (ANOVA) or Kruskal–Wallis followed by post-hoc multiple comparison test was used. Regarding Figure 1F, we exploited the Two-Way ANOVA test. Data, unless otherwise stated, were expressed as means ± sem for the number of cells. The differences were considered significant with *p*-value < 0.05 (one asterisk), <0.01 (double asterisks), <0.005 (triple asterisks) and <0.0001 (quadruple asterisks).

Graphical abstract has been “Created with BioRender.com”.

## 3. Results

### 3.1. Patient Recruitment and Synovial Fluid Collection

Sixteen patients were included in the study. Ten patients had inflammatory arthritis: seven RA (all female, mean age 60 years), and three SpA (1 female, mean age 42 years). Six patients had primary OA (four female, mean age 61 years). The mean VAS (visual analog scale) pain was similar in both groups (5.9 in inflammatory arthritis and 5.8 in OA). The knee was the involved joint in all these patients. Joint inflammation was investigated using Power Doppler (PD) ultrasound (US), to detect effusion, synovial hypertrophy, and synovial PD signal. PDUS and GSUS outcomes have been assessed indicating a comparable level of inflammation among patients (Table 1 and Appendix A); detailed demographic, clinical, psychiatric, and US characteristics of these patients are reported in Table 1.

### 3.2. Characterization of Synovial Fluid-Derived Extracellular Vesicles

Before evaluating the possible role of human-derived EVs in the maintenance of the proper neuronal function, we characterized synovial fluids-derived EVs. The number and the dimension of EVs were assessed by Nanoparticle Tracking Analysis (NTA) using the NanoSight system. Rheumatoid arthritis (RA) and Osteoarthritis (OA) synovial fluids display similar dimensions distribution confirming that the same vesicle population is extracted from OA and RA patients (Figure 1A,B). Importantly, in RA-synovial fluids, we found a greater EVs concentration with respect to what is found in OA-human derivates as revealed by NanoSight analysis (Figure 1A–C); micro BCA protein assay shows a higher protein content in RA- in respect to OA-Evs, even if not significant (Appendix A), and WB analysis of Alix confirmed the presence of EVs (Figure 1D). By biochemical investigation we identified in random OA- and RA- samples, the presence of lipoproteins (HDL), as suggested by the specific marker APOA1; as indicated in Appendix A, the APOA1 signal was lower in synovial fluid samples than in the positive control human serum. Moreover, it was comparable in the RA and OA groups. Finally, OA- and RA-EVs display similar cellular origin as indicated by Fluorescence Activated Cell Sorting (FACS) analysis where we labeled EVs against the following panel of markers: CD177 (for neutrophils origin), CD14 (for monocytes origin), CD62E (as a marker of activated epithelium), CD25 (as a marker of T-reg cells). Results displayed in Figure 1E showed, in general, a greater enrichment of positive EVs in RA than OA with a significant tendency regarding CD25 immunoreactivity, confirming the higher inflammatory cellular component of synovial fluids collected from OA and RA individuals and suggesting a stronger T-reg activation (Figure 1E). We also compared % of EVs population originating from CD177-, CD62-, CD25- and CD14-positive cell types with respect to the total EVs collected from synovial fluids of RA and OA individuals. As indicated in Figure 1F, in the EVs pool extracted from RA synovial fluids we found a higher proportion of CD177^+^-EVs with respect to EVs originating from CD62^+^ and CD25^+^ cells, suggesting a strong neutrophils activation in RA. In OA patients no differences in % were detected between the EVs population originating from CD177-, CD62-, CD25-, and CD14- cells.

### 3.3. EVs Collected from RA and OA Synovial Fluids Lead to Increased Glutamatergic Function but through Opposite Synaptic Changes: Electrophysiological Results

In the literature, it has already been demonstrated that cytokines directly act on neurons modulating neuronal functions [64]. Here, we explored the idea that besides cytokines, also circulating EVs from the periphery may affect neuronal health and transmission, an issue only partially addressed for hematopoietic EVs [42,65]. Interestingly, modifications induced by EVs treatment have been described already, to occur after a short temporal window (within hours) both in vitro and in vivo [40,41,66,67]. Thus, we decided to evaluate the effects mediated by a two hours incubation with EVs extracted from synovial fluid of RA and OA patients on mature hippocampal neurons (14-day-old, DIV 14) derived from wt mouse embryos. To identify a reasonable range of EVs concentration effective in modulating neuronal function, we referred to previous evidence in the literature [40], showing that the minimum amount of MVs able to alter the frequency of spontaneous excitatory transmission corresponds to 2.38 µg/mL; thus we incubated cultures with three different EVs concentrations: 0.75 µg/mL, 2.4 µg/mL, and 4 µg/mL, and analyzed excitatory and inhibitory synaptic function. Through the data obtained during the NTA analysis, we also calculated the total amount of EVs used during the treatment corresponding to each chosen concentration (Appendix A), which coherently increases as the concentration is augmented. Since we did not find any differences in neuronal transmission between cultures treated with the medium (2.4 μg/mL) and higher (4 μg/mL) concentrated EVs, we pulled the results of these two groups all together; thus, we show results obtained with a low (0.75 µg/mL) and medium/high (2.4–4 µg/mL) EVs concentration.

Excitatory and inhibitory postsynaptic currents in miniature (mEPSCs and mIPSCs) have been recorded in DIV14 cultured wild-type hippocampal neurons exposed for two hours to OA- and RA-EVs by patch-clamp technique in whole-cell configuration (in presence of tetrodotoxin (TTX) 1 µM to prevent the generation of action potentials). Frequency and amplitude were analyzed to evaluate potential changes respectively at the presynaptic or postsynaptic compartment. Analysis of miniature events revealed the occurrence of two different changes: one induced by incubating neurons with OA-EVs at medium/high concentrations (2.4 and 4 µg/mL) on excitatory postsynapse, and another generated by RA-EVs at lower amounts (0.75 µg/mL) on inhibitory presynapse.

As shown in Figure 2A–C, no significant differences were detected in mEPSCs frequency upon OA-EVs and RA-EVs delivery, neither at low nor at medium/high concentrations indicating that two hours of treatment does not affect the spontaneous release of glutamate. Regarding the analysis of mEPSCs amplitude, we found significant increments compared to vehicle-treated cultures only in neurons treated with the medium/high concentrated OA-EVs (2.4/4 µg/mL group). These results suggest that EVs derived from synovial fluids extracted from OA patients induce structural changes in the excitatory postsynaptic compartment.

Regarding the inhibitory synaptic transmission (Figure 2A,D,E), incubation of neurons with OA-EVs never modifies excitatory transmission nor in terms of frequency or amplitude. In contrast, exposure of neurons with low concentrated RA-EVs (0.75 µg/mL) induces a significant decrease in mIPSCs frequency compared to the vehicle-treated control cells. The decreased mIPSCs frequency after 2 h incubation with RA-EVs might be due to a lower spontaneous release of neurotransmitters from the presynaptic membrane or reduced GABAergic vesicles. No significant changes were observed in the amplitude of the inhibitory transmission at both concentrations (Figure 2E).

Altogether, these data suggest that EVs derived from the synovial fluids of OA and RA patients may lead to similar changes in terms of global transmission in hippocampal neurons since both favor excitatory transmission but through opposite and specific mechanisms (reduced presynaptic activity/or increased postsynaptic function). Second, different concentrations of EVs modify neuronal function. Indeed, RA-EVs display a greater pathogenic potential, since we observed significant impairments of synaptic function at inhibitory transmission upon exposure to low EVs concentration (0.75 µg/mL); in contrast, modifications induced by OA-EVs treatment are generated by the higher concentrated EVs. Strikingly, no effect was recorded at 2.4 and 4 µg/mL RA-EVs may be due to adaptation mechanisms that may mask a pathological status already established.

### 3.4. EVs Collected from RA and OA Synovial Fluids Lead to Increased Glutamatergic Function but through Opposite Synaptic Changes: Confocal Analysis

We performed immunocytochemical staining on hippocampal neurons treated with OA- and RA-EVs as previously described (2 h incubation at 0.75 µg/mL, 2.4 µg/mL, and 4 µg/mL concentrations). To detect the number of synaptic contacts we analyzed the density of excitatory and inhibitory synapses formed along dendrites exploiting immunoreactivity against vGLUT1 and vGAT proteins, i.e., two vesicular transporters specific for the glutamatergic (major excitatory neurotransmitter) and GABAergic (γ-aminobutyric acid, major inhibitory neurotransmitter) presynapse, respectively. No significant changes were observed in the density of vGLUT1 (Figure 3A–C), neither at low nor at high concentration coherently with functional changes identified by electrophysiology where modifications were more likely restricted to the postsynaptic site. Analyses of vGAT density fully confirm electrophysiological data, indeed neurons incubated for two hours with RA-EVs at 0.75 µg/mL display a significant decrease in vGAT density with respect to controls (Figure 3D,E). No changes resulted with 2.4/4 OA- and RA-EVs (Figure 3F).

In conclusion, specific functional and morphological defects become evident upon exposure of hippocampal neurons to peripheral EVs which cause “per se” an altered network activity since the glial population is largely excluded in our in vitro cultures.

### 3.5. EVs Membranes Contribute to Synaptic Alterations

EVs are cell-derived double layer-limited structures loaded with several components: proteins, lipids, and nucleic acids [34,35]. This means that the above-mentioned effects mediated by EVs exposure may be ascribed to (i) active molecules located at the membrane surface of EVs or to (ii) the material transfer from the EVs′ luminal content to neurons. Interestingly, both these modalities have been described among the EVs′ methods of communication with target cells [40,68]. Thus, to distinguish between these two paradigms of EVs signaling, OA- and RA-EVs were broken by at least three freeze and thaw cycles, pelleted in order to separate and collect soluble components with respect to the pellet membrane. Then, the membrane pellet or the luminal content alone was added to hippocampal neurons at DIV14 and electrophysiological recordings were carried out as in previous experiments.

Analysis of excitatory transmission in neurons treated with OA-broken EVs (membrane pellet) revealed differences in mEPSCs amplitude (Figure 4A,C); no differences were detected upon the incubation with EVS content at high concentration. Exposure of neurons with RA-EVs membrane pellet at low concentration also produced a significant reduction in mIPSCs frequency, whereas RA-EVs content-mediated effects remain undetectable (Figure 4B,D). The amplitude of inhibitory events is not affected either by 0.75 µg/mL RA-EVs content or by the membrane. Finally, to clearly demonstrate that synaptic changes here identified are due to the EVs-membrane component, we repeated our electrophysiological experiments in the presence of a therapeutic agent acting against the transmembrane TNF-alfa (tTNF-alfa), Etanercept 200 ng/mL (ETN). Indeed, TNF-alfa is an inflammatory cytokine significantly enriched in both RA- and OA-EVs [69,70,71] and able to affect both synaptic excitatory and inhibitory transmission [72,73,74].

Neurons received (i) OA- and RA-EVs membrane pellet alone or (ii) a solution composed of EVs membrane + ETN and synaptic activity was evaluated two hours later after drug delivery. As indicated in Figure 4E and G, cultures treated with OA-EVs membrane + ETN display a significant reduction in the mEPSCs amplitude; no effects were detected in terms of mEPSCs frequency. Then, we performed a similar experiment treating neurons with RA-EVs membrane pellets alone or with RA-EVs membrane + ETN. In this case, we did not find any rescued effects upon drug administration (Figure 4F,H), indicating that additional cytokines or EVs membrane components affect the inhibitory transmission. These last data again confirm the specificity of the cellular mechanism induced by OA- and RA-EVs in terms of synaptic changes.

### 3.6. Mood Disorder Severity and EVs Levels in Human Synovial Fluids

Six of sixteen patients (37.5%) reported at least mild psychiatric symptoms as indicated by a CGIs score ≥ 3 as summarized in Figure 5A. The same percentage of subjects reported at least mild anxiety and depressive symptoms, as indicated, respectively, by a HAM-A and a MADRS score ≥ 7. Seven of sixteen patients (42.8%) also showed clinically significant depressive symptoms as assessed by a HAM-D score ≥ 7. Only one patient presented significant autistic traits as shown by an AQ10 score ≥ 6, while prominent neuroticism (calculated as a sum of items 4 and 9 of BFI-10 ≥ 6) was present in eleven of sixteen subjects (68.8% of the total sample). No statistically significant differences were found between the two diagnostic groups (RA + SpA versus OA) in mean rating scale scores (*p* > 0.05), although patients affected by RA or Spa had more severe depressive symptoms (MADRS means scores: 11.70 ± 12.82) with respect to the OA group (MADRS mean scores: 8.00 ± 7.98). Finally, BMI results were directly correlated with depressive symptoms (MADRS total scores: r = 0.58, *p* < 0.05). We decided to correlate EVs synovial concentration of individuals presenting depressed symptomatology and, surprisingly, found a significant enrichment of synovial-derived EVs in patients with mild and moderate depression (Figure 5B).

## 4. Discussion

Data presented in this study lead to three important findings. We demonstrate the direct contribution of peripheral EVs collected from human inflamed periphery to synaptic changes. It is known that in OA and RA conditions, synovial fluids act as a reservoir of EVs and now we display their ability to directly contribute to the onset of synaptic alterations modulating neuronal transmission. Importantly, we identified TNF-alfa as the cytokine that significantly contributes to excitatory synaptic changes, here highlighted. Indeed, we found recovered electrophysiological alterations in neurons treated with the EVs plus the anti-TNF-alfa drug Etanercept. No rescued effects upon Etanercept treatments were detected in terms of inhibitory transmission, indicating once again different mechanisms mediated by EVs on glutamatergic and GABAergic systems.

Interestingly, a high number of clinical evidence indicates that in OA joint tissues, serum, and synovial fluids, TNF-α is more elevated compared with healthy patients [75,76,77]; however, up to now, the use of anti–TNF-α therapeutics in humans is believed more efficacious in improving pain and joint mobility in RA than in the OA condition. Only recently, Etanercept has been administered for 24 weeks, 50 mg/week; thereafter 25 mg/week in a double-blind, randomized, multicenter trial (NTR1192) for one year in patients with symptomatic erosive inflammatory hand osteoarthritis [78]. Here, the authors indicate that in erosive OA, Etanercept was not superior over placebo on VAS pain at 24 weeks. However, in the symptomatic and inflammatory patients completing the study, Etanercept was superior over the placebo both on pain and structural damage assessed by GUSS. Etanercept was especially effective in joints with signs of inflammation.

In a different clinical trial, Etanercept was also found to significantly decrease the severity of fatigue, depression, and anxiety symptoms among patients with psoriasis as indicated in a phase 3 clinical trial [79]. Thus, since our results display rescued neuronal changes upon TNF-alfa inhibition, we speculate that Etanercept may lead to positive psychiatric outcomes also in OA; of course, future preclinical and clinical studies are needed to confirm this point.

Second, the evidence that OA and RA individuals with clinically significant depressive symptoms present higher EVs concentrations with respect to OA and RA patients without depression, suggests the direct correlation between EVs levels and psychiatric comorbidities making peripheral EVs a marker for vulnerability to psychiatric comorbidity.

We are aware that a limitation of the study is the small number of OA and RA patients; however, we achieved a significant increment in EVs concentration in individuals with depressive symptoms. We acknowledge that the limited number of patients in these groups could have masked other significant differences among all patients, as well as the impact of pharmacological therapy followed by each individual in the EVs pool composition. Confirmatory studies in larger sample populations, as well as studies employing different technologies for EV quantification/characterization, are warranted.

Finally, showing opposite changes at the excitatory and inhibitory levels in neurons treated with OA- and RA-EVs, this study lays the scientific basis for personalized medicine in OA and RA patients, hence in the presence of affective symptoms they should be treated with drugs that act through a different mechanism of action on glutamatergic or GABAergic synapse. Importantly, modifications here presented occur already within two hours, a temporal window that even if short, is coherent with previous studies displaying synaptic changes upon exogenous delivery of EVs to neurons [40,80]. Regarding the mechanism of action, our data identify the “EVs membrane” pellet as the principal factor actively involved and responsible for functional modifications here identified.

With regard to psychiatric aspects of the total sample, it is not surprising that at least mild clinically significant affective symptoms are present in about one-third of our patients being affected by inflammatory diseases and pain. This is particularly true for those with RA or SpA as reported in previous investigations by our group on different samples [14,81]. Similarly, increased BMI has been already found to be associated with the onset and severity of depressive symptoms as a result of impaired lipid metabolism [82], weakness of antioxidant defenses [83], and increased inflammation [84]. These findings are consistent with the fact that most patients show prominent neuroticism as this personality trait was associated with depressive disorders [85], and generally poor mental health [54]. On the contrary, previous literature highlighted the diametral opposite nature of autism in comparison with affective disorders, and this could explain the low prevalence of autistic traits in our sample [86].

In conclusion, our study emphasizes the importance of EVs in the context of RA and OA and supports the feasibility of developing strategies that target EVs as potential therapeutic targets to mitigate psychiatric symptomatology.

## Figures and Tables

**Figure 1 cells-11-02276-f001:**
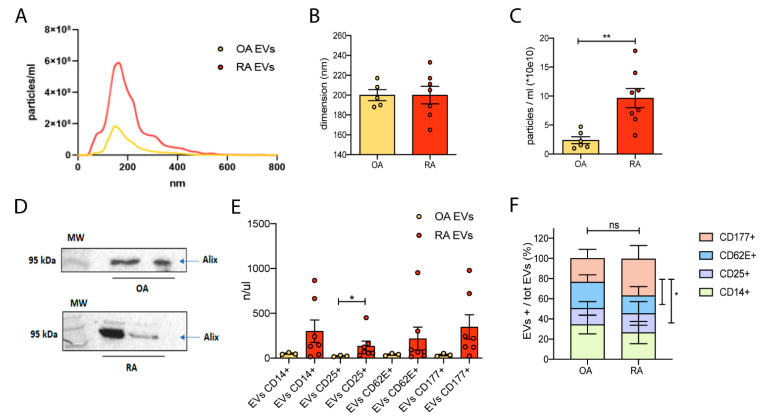
Characterization of synovial fluid-derived extracellular vesicles. (**A**) Representative NanoSight graph: dimension (nm) on x-axis, particles concentration (particles/mL) on y-axis; EVs derived from OA- and RA-synovial fluid are characterized by the same dimension, but a higher RA-particles concentration is obtained from the extraction. (**B**) No difference was observed in the dimension of EVs extracted both from OA- and RA-synovial fluid. (**C**) RA-EVs are characterized by a higher concentration in comparison to OA-EVs. Mann–Whitney test, ** *p* < 0.01. (**D**) Western Blot analysis revealed the presence of Alix in OA- and RA-EVs pellet. (**E**) FACs analysis revealed that OA- and RA-EVs display similar cellular origin; a significant enrichment of CD25-positive EVs was found in RA-derived vesicles, confirming the higher inflammatory cellular component present in synovial fluid collected from RA patients. Mann–Whitney test, * *p* < 0.05. (**F**) EVs originated from distinct cellular populations (CD14^+^, CD25^+^, CD62^+,^ and CD177^+^ cells) and expressed as % of total EVs collected from RA or OA synovial fluids. Two-Way ANOVA (Mixed Model) followed by Tukey’s multiple comparisons test, * *p* < 0.05.

**Figure 2 cells-11-02276-f002:**
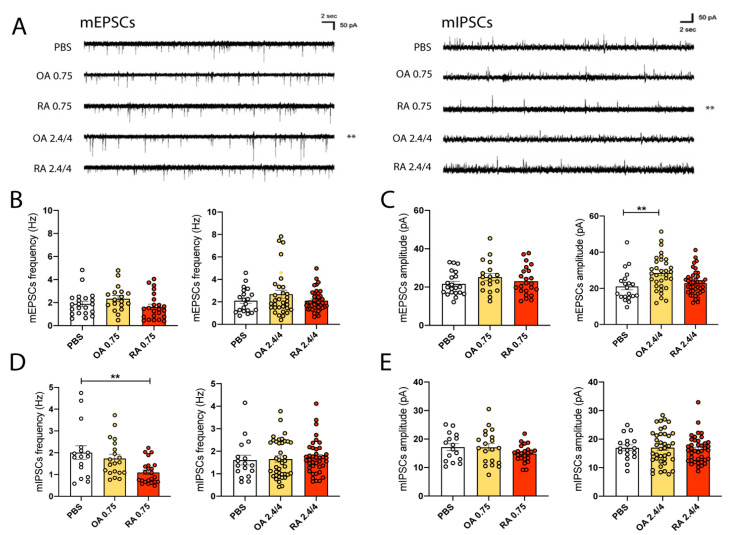
EVs collected from RA and OA synovial fluids lead to increased glutamatergic function but through opposite synaptic changes: electrophysiological results. (**A**) Representative traces of mEPSCs and mIPSCs recorded in 14 DIV hippocampal neurons obtained from wt mouse embryos. (**B**) Analysis of mEPSCs frequency recorded after two hours exposure to OA- and RA-EVs at low (0.75 μg/mL) or medium/high (2.4 and 4 μg/mL) concentration from cultured neurons (low concentration: Kruskal–Wallis test followed by Dunn’s multiple comparison test, ns; number of recorded neurons: PBS = 21, OA = 18, RA = 23; medium/high concentration: Kruskal–Wallis test followed by Dunn’s multiple comparison test, ns; number of recorded neurons: PBS = 20, OA = 32, RA = 41). (**C**) Analysis of mEPSCs amplitude recorded after two hours exposure to OA- and RA-EVs at low (0.75 μg/mL) or medium/high (2.4 and 4 μg/mL) concentration from cultured neurons (low concentration: Ordinary one-way ANOVA followed by Tukey’s multiple comparison test, ns; number of recorded neurons: PBS = 21, OA = 18, RA = 23; medium/high concentration: Kruskal–Wallis test followed by Dunn’s multiple comparison test, ** *p* < 0.01; number of recorded neurons: PBS = 20, OA = 33, RA = 41). (**D**) Analysis of mIPSCs frequency recorded after two hours exposure to OA- and RA-EVs at low (0.75 μg/mL) or medium/high (2.4 and 4 μg/mL) concentration from cultured neurons (low concentration: Ordinary one-way ANOVA followed by Tukey’s multiple comparison test, ** *p* < 0.01; number of recorded neurons: PBS = 16, OA = 20, RA = 23; medium/high concentration: Kruskal–Wallis test followed by Dunn’s multiple comparison test, ns; number of recorded neurons: PBS = 17, OA = 39, RA = 40). (**E**) Analysis of mIPSCs amplitude recorded after two hours exposure to OA- and RA-EVs at low (0.75 μg/mL) or medium/high (2.4 and 4 μg/mL) concentration from cultured neurons (low concentration: Ordinary one-way ANOVA followed by Tukey’s multiple comparison test, ns; number of recorded neurons: PBS = 16, OA = 20, RA = 23; medium/high concentration: Kruskal–Wallis test followed by Dunn’s multiple comparison test, ns; number of recorded neurons: PBS = 17, OA = 39, RA = 40).

**Figure 3 cells-11-02276-f003:**
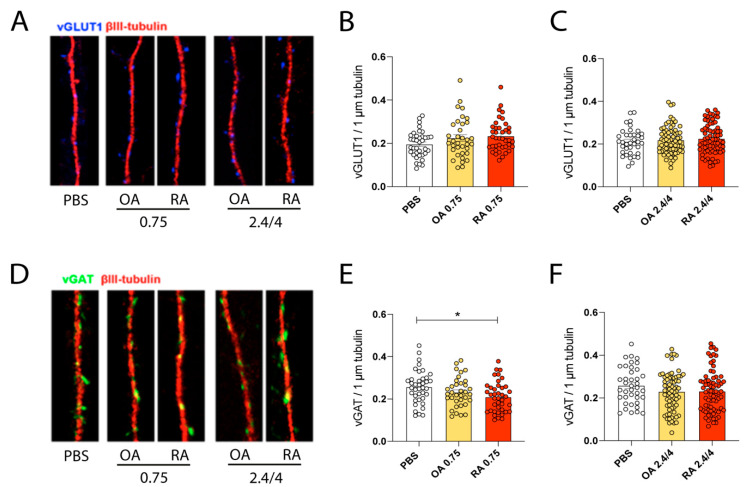
EVs collected from RA and OA synovial fluids lead to increased excitatory transmission but through opposite synaptic changes: confocal analysis. (**A**) Immunocytochemical experiments were performed in wt 14 DIV cultured hippocampal neurons against vGLUT1 (blue) and BIII tubulins (red) after two hours of exposure to OA- and RA-EVs. (**B**,**C**) Analysis of vGLUT1 density shows no differences among PBS and OA- / RA-EVs treated neurons, neither at low nor at medium/high concentration (low concentration: Kruskal–Wallis test followed by Dunn’s multiple comparison test, ns; field analysed for each condition: PBS = 37; OA = 35; RA = 40; medium/high concentration: Ordinary one-way ANOVA followed by Tukey’s multiple comparison test, ns; field analysed for each condition: PBS = 37; OA = 69; RA = 71). (**D**) Immunocytochemical experiments performed in wt 14 DIV cultured hippocampal neurons against vGAT (green) and BIII tubulins (red) after two hours of exposure to OA- and RA-EVs. (**E**,**F**) vGAT density was found increased after exposure to RA-EVs at low concentration; whereas no differences were revealed at medium/high concentration (low concentration: Ordinary one-way ANOVA followed by Tukey’s multiple comparison test, * *p* < 0.05; field analysed for each condition: PBS = 39; OA = 35; RA = 40; medium/high concentration: Kruskal–Wallis test followed by Dunn’s multiple comparison test, ns; field analyzed for each condition: PBS = 39; OA = 71; RA = 74).

**Figure 4 cells-11-02276-f004:**
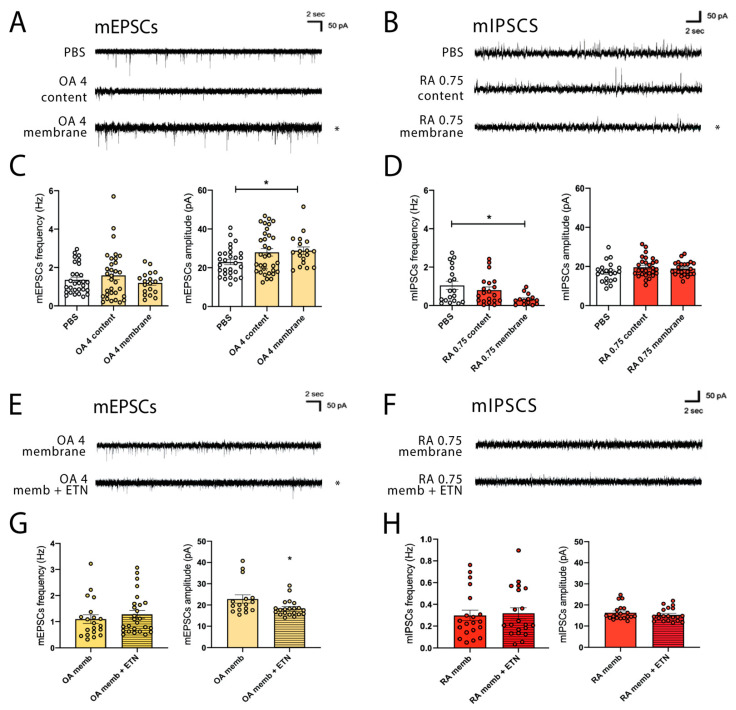
EVs membranes contribute to synaptic alterations. (**A**) Representative traces of mEPSCs recorded in 14 DIV hippocampal neurons obtain after two hours of exposure to OA-EVs content and membrane at high concentration. (**B**) Representative traces of mIPSCs recorded in 14 DIV hippocampal neurons obtain after two hours of exposure to RA-EVs content and membrane at low concentration. (**C**) Analysis of mEPSCs frequency and amplitude recorded after two hours of exposure to OA-EVs content and membrane at high concentration from cultured neurons. No differences in terms of frequency were detected, whereas OA-EVs membranes increase mEPSCs amplitude (frequency: Kruskal–Wallis test followed by Dunn’s multiple comparison test, ns; number of recorded neurons: PBS = 29, OA content = 32, OA membrane = 19; amplitude: Kruskal–Wallis test followed by Dunn’s multiple comparison test, * *p* < 0.05; number of recorded neurons: PBS = 29, OA content = 32, OA membrane = 18). (**D**) Analysis of mIPSCs frequency and amplitude recorded after two hours of exposure to RA-EVs content and membrane at low concentration from cultured neurons. No differences in terms of amplitude were detected, whereas RA-EVs membranes decrease mIPSCs frequency (frequency: Kruskal–Wallis test followed by Dunn’s multiple comparison test, * *p* < 0.05; number of recorded neurons: PBS = 19, RA content = 20, RA membrane = 14; amplitude: Ordinary one-way ANOVA followed by Holm–Sidak’s multiple comparison test, ns; number of recorded neurons: PBS = 21, RA content = 28, RA membrane = 22). (**E**) Representative traces of mEPSCs recorded in 14 DIV hippocampal neurons obtain after two hours of exposure to OA-EVs membrane (high concentration) and OA-EVs membrane (high concentration) + ETN. (**F**) Representative traces of mIPSCs recorded in 14 DIV hippocampal neurons obtain after two hours of exposure to RA-EVs membrane (low concentration) and RA-EVs membrane (low concentration) + ETN. (**G**) Analysis of mEPSCS frequency and amplitude recorded after two hours of exposure to OA-EVs membrane (high concentration) + ETN from cultured neurons. No differences in terms of frequency were detected, whereas ETN treatment reduces mEPSCs amplitude (frequency: Mann–Whitney test, ns; number of recorded neurons: OA membrane = 20, OA membrane + ETN = 26; amplitude: Mann-Whitney test, * *p* < 0.05; number of recorded neurons: OA membrane = 16, OA membrane + ETN = 21). (**H**) Analysis of mIPSCs frequency and amplitude recorded after two hours of exposure to RA-EVs membrane (low concentration) + ETN from cultured neurons. No differences in terms of both frequency and amplitude were detected upon ETN treatment (frequency: Mann-Whitney test, ns; number of recorded neurons: RA membrane = 20, RA membrane + ETN = 20; amplitude: Mann-Whitney test, ns; number of recorded neurons: RA membrane = 20, RA membrane + ETN = 20).

**Figure 5 cells-11-02276-f005:**
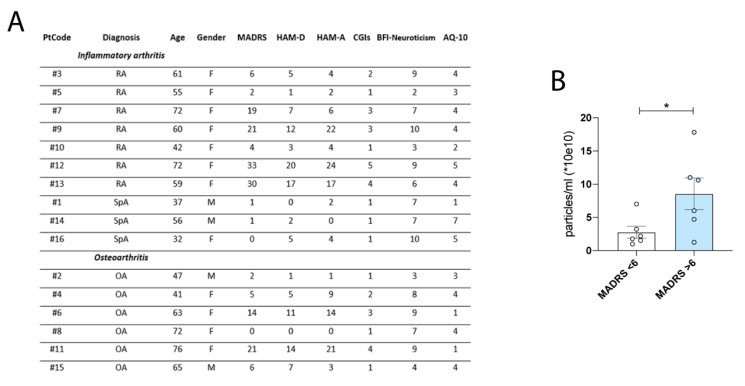
Mood disorder severity and EVs levels in human synovial fluids. (**A**) Table summarizing age, gender, and psychiatric scales (MADRS, HAM-D, HAM-A, CGI, BFI-Neuroticism, and AQ-10) for each RA and OA patient. 37.5% of the total patients report at least mild psychiatric symptoms, mild anxiety, and depressive symptoms. AQ-10: 10-item Autism Spectrum Quotient; BFI: 10-item Big Five Inventory, Neuroticism is calculated as the sum of item 4 and 9; CGIs: Clinical Global Impression-Severity of Illness; HAM-A: Hamilton Anxiety Rating Scale; HAM-D: Hamilton Depression Rating Scale; MADRS: Montgomery Asberg Depression Rating Scale; OA: osteoarthritis; RA: rheumatoid arthritis; SpA: spondyloarthritis. (**B**) EVs concentration is higher in patients presenting mild and moderate depressive symptoms (MADRS > 6; Unpaired *t*-test, * *p* < 0.05).

**Table 1 cells-11-02276-t001:** Table summarizing demographic, clinical, and ultrasonographic characteristics of all the sixteen patients included in the study. Even though ten patients were affected by inflammatory arthritis (seven RA and three SpA) and six patients had primary OA, a comparable level of inflammation was detected among all of them.

Pt Code	Diagnosis	Age	Sex	RF/ACPA	VAS Pain	US Grading	
Inflammatory Arthritis						GSUS	PDUS
#3	RA	61	F	neg/neg	5	2	1
#5	RA	55	F	pos/pos	8	1	0
#7	RA	72	F	neg/neg	6	1	0
#9	RA	60	F	pos/pos	10	2	3
#10	RA	42	F	neg/neg	7	2	3
#12	RA	72	F	neg/neg	9	1	0
#13	RA	59	F	neg/neg	10	2	1
#1	SpA	37	M	neg/neg	3	1	2
#14	SpA	56	M	neg/neg	0	3	3
#16	SpA	32	F	neg/neg	1	3	3
**Osteoarthritis**							
#2	OA	47	M	-	4	0	0
#4	OA	41	F	-	5	2	0
#6	OA	63	F	-	7	2	1
#8	OA	72	F	-	4	1	0
#11	OA	76	F	-	8	2	2
#15	OA	65	M	-	7	2	3

RA: rheumatoid arthritis; OA: osteoarthritis; SpA: spondyloartjritis; RF: rheumatoid factor; ACPA: anti-citrullinated protein antibody; VAS: visual analog scale; US ultrasonography; GSUS: gray scale US; PDUS: power Doppler US.

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
