# Peer review of "Synovial Fluid-Derived Extracellular Vesicles of Patients with Arthritides Contribute to Hippocampal Synaptic Dysfunctions and Increase with Mood Disorders Severity in Humans"

_cells, 2022, doi:10.3390/cells11152276_

Round 1
Reviewer 1 Report
In this paper, the authors investigated whether inflammatory EVs present in the synovial fluid of patients with arthritides may play a role in the psychiatric symptoms frequently present in patients with osteoarthritis (OA) or rheumatoid arthritis (RA). Although the number of patients is quite low, the authors convincingly show that patients with depressive symptoms exhibit higher EV concentration in synovial fluids, and that EVs from the synovial fluids directly influence neurotransmission of hippocampal neurons in primary cultures. Specifically, EVs from OA patients increase mEPSC amplitude while those of RA patients decrease mIPSC frequency, both effects leading to potentiation of excitatory transmission. These results increase current knowledge on mechanisms by which peripheral inflammation may influence brain function, a topic of great interest for both basic scientists and clinicians.
Overall, experiments are clearly described and appropriate controls included. The reference are appropriate.
Below my comments and suggestions to strenghthen the paper
-Abstract: the quality of English should be improved (see for example Line 27-28)
-WB in fig1D, besides the EV marker Alix, other EV markers should be shown as well as EV negative markers, to exclude the presence of EV contaminants, according to MISEV guidelines
-Fig 1E. It would be informative to show the relative percentage of EVs of different cell origin, in order to understand whether there are changes in EV production from distinct cell types between RA and OA patients
- Given that the bioactive molecules able to modulate neurotransmission are surface membrane components of EVs, I wonder whether transmembrane TNF may pay a role in enhancement of mEPSC amplitude upon incubation with OA EVs. This hypothesis can be easily tested by the use TNF decoy receptors
Reviewer 2 Report
This is an interesting idea to test the effect of extracellular vesicles obtained for synovial fluids from (osteo)arthritis patients on excitatory synaptic transmission. A poor mental health in these patients is recognised and a serious problem but the underlying mechanism is poorly understood and the molecules involved not well studied. Extracellular vesicles (EVs) are potential candidates and this study showed the first evidence that EVs from synovial fluids of arthritis patients can evoke an effect on neurons, but the data must be interpreted with caution. Although I like challenging and provocative titles, in this title the word MAY between arthritides and contribute is right here.
Overall comments:
1. It is unclear if these arthritis patients have received treatment before and what the duration of their disease is and how this may confound the results. Only 2 of the 7 RA patients are RF and ACPA positive, a hallmark of RA, so based on which clinical and laboratory parameters are the RF/ACPA negative patients diagnosed as RA?
2. The purity of the EVs is a major concern. I calculated that the amount of protein is between 115-120 fg/particle and this is to our estimate too high and should be around 10 fold less. This is due to the technigue used to isolate these EVs as a single ultracentrifuge step is not enough to obtain pure EVs. This means that protein aggregates (immunecomplexes) or lipoprotein particles might contribute to the observed effect. This may also explain why the EV membranes are the active component as both aggregates and particles are not destroyed but freeze-thaw cycles. For this also a WB showing non-EV proteins/plasma proteins should be included to convincingly proof the purity of EVs.
3) EV analysis is correct but this fullfils only the minimal requirements. More EV markers should be analyzed and morphology should be checked with cryoTEM. The NTA analysis as presented in fig 1a (peak between 1500-2000nm) and 1b (200 nm, mean or mode?) seems not to correspond.
4) Data presentation: Firstly, there is no EV control for the electrophysical results. It is interpreted that the observed effects are due to EVs (I have my doubt due to the purity) and specific for synovial fluid, yet EVs taken from blood of the same patients or synovial fluid EVS form trauma patients have not been included. So the evidence is suggestive and not conclusive. Hence may comments on the title. Secondly, the number of measurements in figures 2-4 is far higher than the number of subjects included in this study. So what do you represent here, the number of measurements, experiments, samples analyzed? This all has to do with the statistical power, and also for that reason I don't agree that the portein concentrations of 2,4 and 4 ug are pulled as this articifially increases the numbers and suppl fig 2 shows that at least on particle numbers these tow dosages are significantly different. Thirdly, fig 5 is interesting but also not truly informative. It suggest that a higher amount of synovial EVs correspond to more depression regardless of the disease. This could be an interesting finding but would be stronger if it was compared/correlated to the significant effects as presented in figures 3 and 4.
5) In comparison to the introduction (rather long) the discussion is short and on the mechanism is not given any speculation. Also the remark on BMI triggered my critique on the purity as VLDL and chylomicron particles are in the size ranges as shown in figure 1a. The discussion should also inlcude a paragrapgh on the weaknesses of the study which are: small number of subject, no proper control, a species difference in the in vitro experiments, and maybe more.
Round 2
Reviewer 1 Report
The authors successfully addressed the issues raised in my review
New electrophysiological data with etarnecept have increased the study relevance